# A Comprehensive Narrative Review on the History, Current Landscape, and Future Directions of Hepatocellular Carcinoma (HCC) Systemic Therapy

**DOI:** 10.3390/cancers15092506

**Published:** 2023-04-27

**Authors:** Alexander Lazzaro, Kevan L. Hartshorn

**Affiliations:** 1Department of Medicine, Boston Medical Center, Boston, MA 02118, USA; 2Section of Hematology Oncology, Boston University Chobanian and Avedisian School of Medicine, Boston Medical Center, Boston, MA 02118, USA; khartsho@bu.edu

**Keywords:** hepatocellular carcinoma (HCC), liver, hepatology, cancer, systemic therapy, tyrosine kinase inhibitor, immune checkpoint inhibitor, monoclonal antibody

## Abstract

**Simple Summary:**

Hepatic and intrahepatic bile duct cancer is the sixth most frequently diagnosed cancer in the world. A total of 830,200 people died from liver cancer globally in 2020. It is estimated that hepatocellular carcinoma (HCC), an aggressive, primary malignant liver tumor, accounts for approximately 85% of all primary liver tumors. Despite the improvements in surveillance screening programs, the prevalence of HCC is expected to rise significantly over the coming decades. Beginning with the approval of sorafenib in 2007, there has been a monumental amount of progress attained in the management of advanced HCC with systemic therapies. Certain major gaps in therapy remain. This narrative review details the history, current landscape, and future directions of HCC systemic therapy. We hope this review will heighten interest in the field of HCC systemic therapies, provide a clear outline of the current data and strategy for treatment, and sensitize readers to new developments that are likely to emerge.

**Abstract:**

We provide a comprehensive review of current approved systemic treatment strategies for advanced hepatocellular carcinoma (HCC), starting with the phase III clinical trial of sorafenib which was the first to definitively show a survival benefit. After this trial, there was an initial period of little progress. However, in recent years, an explosion of new agents and combinations of agents has resulted in a markedly improved outlook for patients. We then provide the authors’ current approach to therapy, i.e., “How We Treat HCC”. Promising future directions and important gaps in therapy that persist are finally reviewed. HCC is a highly prevalent cancer worldwide and the incidence is growing due not only to alcoholism, hepatitis B and C, but also to steatohepatitis. HCC, like renal cell carcinoma and melanoma, is a cancer largely resistant to chemotherapy but the advent of anti-angiogenic, targeted and immune therapies have improved survival for all of these cancers. We hope this review will heighten interest in the field of HCC therapies, provide a clear outline of the current data and strategy for treatment, and sensitize readers to new developments that are likely to emerge in the near future.

## 1. Introduction

Hepatic and intrahepatic bile duct cancer is the sixth most frequently diagnosed cancer in the world. A total of 905,700 people were diagnosed with it and 830,200 people died from liver cancer globally in 2020 [1]. In 46 countries, liver cancer was among the top three causes of cancer-related death. Liver cancer is the second-most lethal primary cancer after pancreatic cancer, with a five-year survival rate of approximately 18 percent [1]. It is estimated that hepatocellular carcinoma (HCC), an aggressive, primary malignant liver tumor that usually occurs in the setting of chronic liver disease, particularly in patients with cirrhosis or chronic hepatitis B virus, accounts for approximately 85% of all primary liver tumors with cholangiocarcinoma compromising the majority of remaining cases [2].

Using age-standardized incidence and mortality rates (ASRs) per 100,000 person years calculated from data extracted from the 2020 GLOBOCAN database, the number of new cases of liver cancer per year is predicted to increase by 55.0% between 2020 and 2040 with a possible 1.4 million people being diagnosed in 2040 [1]. The worldwide incidence and trends of HCC vary according to global geographic location with differences likely due to regional variations in exposure to hepatitis viruses and environmental exposures as well as vaccination, screening, and treatment options.

Despite the evidence of a survival benefit with the increased surveillance screening of HCC, over 60% of HCC in clinical practice is diagnosed at late stages, suggesting both inadequacies in the surveillance process and the necessity for efficacious downstream therapeutic options [3,4]. Current clinical guidelines for the standard of HCC care include potentially curative therapies (i.e., radiofrequency or microwave ablation, resection, and liver transplantation) for early tumors, transarterial chemoembolization (TACE) for intermediate-stage tumors, and various systemic drugs for advanced tumors in the first and second lines [5]. As of 2020, the median survival time is beyond 5 years (>6 years for resection/ablation, and 10 years for transplantation) for early stages, approximately 20–30 months for intermediate stages, and 10–16 months for advanced-stage HCC [5,6]. There has been a remarkable amount of research in the past 15 years towards addressing improvements in systemic therapy for advanced-stage HCC. 

The HCC treatment algorithm guides treatment decision making based on the extent of the primary lesion, performance status, vascular invasion, extrahepatic spread, and baseline liver function as is evident in the Barcelona Clinic Liver Cancer (BCLC) staging system [7]. There are some concerns regarding the universal applicability of the BCLC algorithm [7]. Currently, debate exists regarding the use of tumor extent alone to guide TNM staging, the un-validated use of the Child–Pugh classification to stratify prognosis according to underlying liver function in patients with HCC, and pertinently for this review, the advent of new, rapidly developing treatment modalities, especially for systemic therapy, that vary tremendously according to availability in resource-poor settings, up-to-date knowledge, and expertise. Broadly speaking, patients who are not candidates for resection because of tumor extent or underlying liver dysfunction can be screened for liver transplantation, locoregional liver-directed therapies, and systemic therapy. This narrative review will focus on all elements pertaining to HCC that require systemic therapy. 

## 2. Evolution of Systemic Therapy

Prior to the SHARP trial and approval of sorafenib, systemic therapy had not been used routinely for patients with advanced HCC for a myriad of reasons including but not limited to HCC’s high rate of expression of drug resistance genes and difficulties with chemo-toleration owing to underlying hepatic dysfunction [8]. Ablative approaches including transarterial chemoembolization or palliative systemic therapy with conventional cytotoxic chemotherapy agents were variably used [9]. The use of sorafenib was based on a recognition of the important roles of proangiogenic factors, vascular endothelial growth factor (VEGF), the platelet-derived growth factor (PDGF), and the fibroblast growth factor (FGF) in neovascularization, invasiveness, and metastatic potential. Since the demonstration of overall survival (OS) benefits of sorafenib, there has been an explosion of systemic therapy clinical trials for HCC, some of which have failed to meet or improve the survival benefits of sorafenib. However, more recent trials, particularly those incorporating immunotherapy, have shown improved survival compared with sorafenib. A chronological timeline of such HCC systemic therapy clinical trials, including those that resulted in the approval of agents as first- and second-line options as well as relevant ones that failed along the way, is illustrated in Figure 1. 

Currently, systemic therapy is appropriate for those HCC patients who are not amenable to curative or locoregional therapy and have adequate performance status and underlying liver function. This includes HCC patients with extrahepatic spread, with tumors confined to the liver which have progressed after locoregional therapies, extensive vascular invasion, or a large intrahepatic tumor burden unsuitable for locoregional approaches. Lastly, prior to initiating systemic treatment, it is important to note the parameters for viral hepatitis screening as well as the predictive markers of response. In the former, a 2020 provisional clinical opinion from the American Society of Clinical Oncology (ASCO), which endorses universal HBV screening for all patients beginning systemic anti-cancer therapy (cytotoxic chemotherapy, immunotherapy, and molecularly targeted therapy) using HbSAg, anti-HBc, IgG, anti-HBs, and less compelling HCV screening recommendations prior to initiating potentially immunosuppressive chemotherapy, is supported [10]. In addition to overall survival (OS; the principally used primary endpoint in cancer research), other potential surrogate endpoints, such as response rate and progression-free survival (PFS), are also currently in use [11]. To validate and provide a common framework for assessing treatment response in clinical trials on HCC, first the RECIST (the Response Evaluation Criteria in Solid Tumors) and then the modified RECIST (mRECIST) guidelines, which mainly measure tumor viability, were established [11]. In the case of sorafenib, survival extension occurred despite the low objective response rate (ORR) assessed by standard RECIST criteria.

### 2.1. Systemic Therapy as First-Line Therapy

#### 2.1.1. For HCC Patients with Preserved Liver Function and Functional Status

Systemic therapy as a first-line option is appropriate for patients with advanced, unresectable HCC who are unsuitable for locoregional therapy and whose liver function is adequate enough to tolerate therapy (i.e., Child–Pugh class A or B cirrhosis). The recommended approach is dependent firstly upon active ongoing clinical trials testing new therapeutic strategies. 

Currently, for healthy patients with an ECOG (Eastern Cooperative Oncology Group) performance status of 0 or 1, no worse than Child–Pugh class A cirrhosis, tumors that have not recurred following liver transplantation, and those following the management of esophageal varices, the suggested first-line therapy options are outlined below and summarized in Table 1. Notably, the selection criteria and stratification factors just described, were established by the benchmark for clinical trial design for HCC, the previously mentioned SHARP study. 

#### 2.1.2. Sorafenib 

Until 2007, no effective systemic therapy existed that improved survival for patients with advanced HCC [9]. This changed with preliminary studies suggesting that sorafenib, an oral multikinase inhibitor of serine-threonine kinases Raf-1 and B-Raf and the receptor tyrosine kinase inhibitor (TKI) of VEGFRs-1, 2, and 3 and platelet-derived growth factor receptor beta (PDGFR-beta), may be therapeutically effective for HCC given the kinases’ role in cellular signaling implicated in the molecular pathogenesis of HCC [12]. In preclinical experiments, sorafenib demonstrated anti-proliferative activity in liver cancer cell lines, reduction in tumor angiogenesis and tumor-cell signaling, and an increase in tumor cell apoptosis in a mouse xenograft model of human HCC [13]. Eventual experimentation and publication of the multicenter, phase III, double-blind, placebo-controlled SHARP trial examined the primary outcomes of OS and the time to symptomatic progression. Median overall survival (mOS) was 10.7 months in the sorafenib group and 7.9 months in the placebo group (Hazard ratio (HR) in the sorafenib group, 0.69; 95% confidence interval (CI), 0.55 to 0.87; *p* < 0.001) and thus efficacy was established for sorafenib over a placebo as a first-line systemic therapy option in unresectable, advanced HCC [8]. Notably, the overall incidence of treatment-related adverse events was 80% in the sorafenib group and 52% in the placebo group. Most commonly, the adverse events reported were predominantly grade 1 or 2 in severity and GI (diarrhea), constitutional (weight loss), or dermatological (hand-foot-skin reaction, alopecia, etc.) in nature [8]. Nonetheless, sorafenib was approved by the US FDA in November 2007 for advanced HCC as the first-line standard-of-care treatment. An additional trial confirmed the survival benefit of sorafenib in an Asian-Pacific population [14]. 

#### 2.1.3. Unsuccessful TKI Challengers of Sorafenib 

Since the groundbreaking success of sorafenib in the SHARP trial, several further phase III clinical trials (up until 2018) failed to demonstrate equivalence with sorafenib as first-line advanced HCC treatments. This includes brivanib, a multitargeted TKI of VEGFRs and fibroblast growth factor receptors (FGFRs), in the BRISK-FL study which did not meet its primary endpoint of OS non-inferiority versus sorafenib; sunitinib, a multitargeted TKI of PDGF-Rs, VEGFRs, RET, G-CSF-R, FLT-3 and c-KIT, which was significantly inferior to sorafenib with more frequent and severe toxicity; linifanib, a multitargeted TKI of VEGFRs and PDGFRs, which did not meet the predefined superiority and non-inferiority OS primary endpoints and had a worsened safety profile [15,16,17].

#### 2.1.4. Unsuccessful Combination Therapy: Sorafenib and Doxorubicin 

In addition, studies investigated the combination of chemotherapy agents with sorafenib in efforts to demonstrate improved clinical benefits as a first line. Notably, doxorubicin, an anthracycline drug that previously showed promise with feasibility and tolerability in a phase I trial and then a significant improvement in OS in a randomized, double-blind, phase II study when joined with sorafenib, was further explored as a potential pairing with sorafenib [18,19]. The hypotheses at the time suggested a possible synergism between the two via the inhibition of the Ras/Raf/MEK/ERK pathway preventing the activation of the multidrug resistance pathway and additive effects via anthracyclines’ modulation of angiogenesis [20,21]. Ultimately, however, the CALGB 80802 phase III clinical trial studying sorafenib and doxorubicin compared to sorafenib monotherapy was halted after the accrual of 356 (out of planned 480) patients with a futility boundary crossed at a planned interim analysis [22]. Overall, the primary endpoint of mOS was 9.3 months (95% CI, 7.3–10.8 months) in the doxorubicin and sorafenib arm and 9.4 months (95% CI, 7.3–12.9 months) in the sorafenib alone arm (HR, 1.05; 95% CI, 0.83–1.31). In addition, hematologic toxicities, especially grade 3 or 4 neutropenia (36.8%) and thrombocytopenia (17.5% vs. 2.4%) occurred significantly more frequently in the combination group [22]. The discrepancy between the phase II and III trials could in part be explained by the use of doxorubicin in lieu of sorafenib as the control for the CALGB phase II trial [22,23]. 

#### 2.1.5. Unsuccessful Combination Therapy: Sorafenib and an EGFR Inhibitor

Next, in the SEARCH phase III clinical trial, erlotinib, a TKI of EGFR (endothelial growth factor receptor), was combined with sorafenib and compared against sorafenib monotherapy in the first-line systemic therapy setting. It was hypothesized that EGFR inhibition, via erlotinib, would enhance tumor response due to both the implicated roles of the EGFR pathway in the pathogenesis of HCC and that of EGFR activation interfering with HCC response to sorafenib [24,25]. A phase I trial of sorafenib and erlotinib in patients with advanced solid tumors revealed good toleration and no pharmacokinetic interactions [26]. Ultimately, in the SEARCH trial, there was no improvement of OS or PFS in patients with unresectable HCC compared to sorafenib monotherapy [27]. 

#### 2.1.6. Lenvatinib, the First Approved Alternative to Sorafenib 

Finally, studies conducted on lenvatinib, an inhibitor of VEGF receptors 1–3, FGF receptors 1–4, PDGF receptors alpha, RET, and KIT revealed promising early results. Its molecular nature was smaller than that of sorafenib with more potent activity against VEGF receptors and the FGF family. In the open-label, phase III, multicenter, non-inferiority, REFLECT clinical trial, OS as a primary endpoint was compared between unresectable HCC patients receiving sorafenib vs. lenvatinib. The median survival time for lenvatinib of 13.6 months (95% CI, 12.1–14.9) was non-inferior to that of sorafenib (12.3 months, 10.4–13.9; HR, 0.92; 95% CI 0.79–1.06) [28]. Compared to sorafenib, similar rates and types of adverse events were reported with the exception of increased reports of grade 3- or 4-associated hypertension (23 vs. 14 percent) and proteinuria in the lenvatinib group [28]. In contrast, lenvatinib appeared to have less hand–foot syndrome than sorafenib did. Consequently, US FDA approval was granted in August 2018, thus marking the first alternative to sorafenib (albeit not superior to it) for the first-line treatment of unresectable HCC, with NCCN (National Comprehensive Cancer Network) guidelines suggesting limiting its use to individuals with no worse than Child-Pugh class A cirrhosis. 

#### 2.1.7. Bevacizumab and the Emergence of Anti-VEGF Monoclonal Antibodies 

Throughout this time interval, studies were underway on monoclonal antibodies directed against VEGF, in similar hopes of reducing tumor vascularity and growth. Bevacizumab is a humanized monoclonal antibody that binds to VEGF-A [29]. Various initial experiments and subsequent phase II clinical trials were conducted that investigated bevacizumab first in vitro, then in mouse models and then as either monotherapy, combination therapy with chemotherapy (capecitabine, oxaliplatin, and gemcitabine), or combination therapy with EGFR inhibitors in efforts to demonstrate its efficacy as an alternative first-line systemic therapy option for unresectable HCC [30,31,32,33,34,35,36,37]. A systematic review of the efficacy and safety of bevacizumab for the treatment of advanced HCC was carried out, which included 8 trials and 300 patients with bevacizumab given in various regimens, as mentioned above. Ultimately, the drug showed promise as an effective and tolerable agent that compared favorably to sorafenib and warranted comprehensive examination in the phase III setting [38]. 

#### 2.1.8. Immune Checkpoint Inhibitors (ICIs): Origin in Treatment for Advanced HCC 

Concomitantly, immune checkpoint inhibitor immunotherapy research was underway to find alternative treatment modalities for cancer research at large, including that for unresectable HCC. The premise is based on the dual understanding that the development of cancers is a multi-step process, that is in short characterized by the accumulation of genetic and epigenetic alterations that drive or reflect tumor progression, in addition to the fact that developing cancer cells become distinguished from their normal counterparts yet are rarely rejected spontaneously, reflecting their ability to maintain an immunosuppressive microenvironment [39]. Notably, programmed death ligand 1 (PD-L1, i.e., B7-H1) is noted to be expressed on many cancer and immune cells and by binding to programmed death-1 (PD-1) and CD80, negative regulators of T-lymphocyte activation, it suppresses T-cell migration, proliferation, and secretion of cytotoxic mediators, thereby restricting tumor cell killing and blocking the cancer immunity cycle [40]. Hence, extensive research has been underway to block the binding of PD-L1 to its receptors, enhancing anti-cancer immunity, with hopes of developing efficacious advanced HCC treatment options. 

Despite the clear clinical benefits achieved with ICI therapy thus far, it is worth noting their use has been associated with a new spectrum of side effects, related to their mechanism of action, which is distinct from that of other systemic therapies such as cytotoxic chemotherapy [41]. Notably, their side effects predominantly effect the GI, dermatologic, hepatic, endocrine, and pulmonary systems though any bodily system may be affected. The incidence and onset of immune-related adverse effects has been shown to depend on the type of cancer, the class and dose of ICI used, and specific factors related to the patient [41]. Because HCC by and large develops within a background of cirrhosis, which on its own leads to systemic manifestations, the potential for harmful risks could not be ignored. However, in general, studies have shown that metrics for immune related adverse events are not significantly higher for HCC treatment than they are for other cancers [42]. Certain patient populations with specific co-morbidities must be more carefully examined when undergoing ICI therapy, as is evident in those with IBD (inflammatory bowel disease) undergoing disease reactivation at higher rates [43]. 

Initial immune checkpoint inhibitor studies on HCC were developed upon the success highlighted in melanoma research seen in the form of pembrolizumab treatment, in the Keynote-001 study, and nivolumab, in the CheckMate 066 study—both monoclonal antibodies directed against PD-1 [44,45]. When adapted for advanced first and second-line HCC treatment there were mixed results. In an updated report of the second cohort of patients (51 in total) who had not received any prior systemic therapy for advanced HCC in the Keynote-224 phase II trial, an objective response rate (ORR) of 16% and OS of 17 months with a consistent safety profile was observed in patients receiving pembrolizumab [46]. In the CheckMate 459 study, a randomized, multicenter, open-label, phase III study investigating nivolumab monotherapy vs. sorafenib monotherapy, a mOS of 16.4 months (95% CI, 13.9–18.4) compared with the 14.7 months (95% CI, 11.9–17.2) for sorafenib (HR, 0.85; 95% CI, 0.72–1.02; *p* = 0.075) was demonstrated [47]. Although nivolumab did not significantly improve OS compared to sorafenib, clinical activity and a favorable safety profile were observed, thus leading to recommendations for its use to patients in whom TKIs and anti-angiogenic drugs are contraindicated or have substantial risks.

In slightly earlier ICI studies, the role of CTLA-4 (cytotoxic T-lymphocyte associated protein-4)-directed monoclonal antibodies was explored following the success of ipilimumab against melanoma in 2011 [48]. With the understanding that CTLA-4 is an inhibitory co-receptor that interferes with T-cell activation and proliferation, a pilot phase I study of tremelimumab, an anti-CTLA-4 monoclonal antibody, was conducted and demonstrated a good safety profile with a partial response rate of 17.6% amongst 21 enrolled patients [49]. Thus, further studies were warranted.

#### 2.1.9. Combination Therapy: Atezolizumab and Bevacizumab

Combination studies were conducted to explore the additive effects of ICIs and anti-VEGF therapies. A phase Ib study of atezolizumab, a monoclonal antibody checkpoint inhibitor selectively targeting PD-L1, and bevacizumab was conducted in patients with unresectable HCC. It demonstrated an ORR of 36%, a median progression-free survival (mPFS) advantage of 5.6 vs. 3.4 months, and an acceptable side-effect profile [50]. In the follow-up, global, open-label, phase III IMbrave150 trial, patients with unresectable HCC who had not previously received systemic treatment were randomized to receive either bevacizumab and atezolizumab or sorafenib until unacceptable toxic effects occurred or there was a loss of clinical benefit. The primary endpoints were OS and PFS. In the intention-to-treat population, OS at 12 months was 67.2% (95% CI, 0.42 to 0.79; *p* < 0.001) with atezolizumab–bevacizumab and 54.6% (95% CI, 45.2 to 64.0) with sorafenib. The median PFS was 6.8 months (95% CI, 5.7 to 8.3) and 4.3 months (95% CI, 4.0 to 5.6) in the respective groups (HR for disease progression or death, 0.59; 95% CI, 0.47 to 0.76; *p* < 0.001). In conclusion, when compared head-to-head, in patients with unresectable HCC, atezolizumab combined with bevacizumab resulted in better overall and progression-free survival outcomes than did sorafenib, a previously preferred systemic therapy first-line option [51]. An updated OS analysis following an additional 12 months of follow up was revealed at the 2021 ASCO Gastrointestinal Cancer Symposium and demonstrated a sustained clinical efficacy benefit with atezolizumab + bevacizumab vs. sorafenib with a mean OS of 19.2 months vs. 13.4 months (HR, 0.66; 95% CI, 0.52–0.85; *p* < 0.0009) [52]. Thus, sorafenib was firmly, for the first time, supplanted as the recommended first-line systemic therapy option for advanced HCC, with the exception being patients with untreated varices or those with contraindications for VEGF inhibitors or immunotherapy. 

#### 2.1.10. Combination Therapy: Sintilimab and Bevacizumab

Next, the Chinese ORIENT-32 Trial was published in 2021 with an investigation into the combination of sintilimab, an alternate anti-PD-1 monoclonal antibody, and a bevacizumab biosimilar (IBI305). In this randomized, open-label, phase II-III study, the combination was compared against sorafenib monotherapy in unresectable HCC [53]. Similarly, this combination regimen showed a significant OS and PFS benefit versus sorafenib monotherapy in the first-line setting for unresectable (exclusively HBV-associated) HCC, with an acceptable safety profile. To date, this regimen remains under regulatory review and does not yet have FDA approval as an alternative systemic HCC therapy option. 

#### 2.1.11. Combination Therapy: Tremelimumab and Durvalumab 

Results from the HIMALAYA phase III clinical trial, a study investigating the combination therapy of tremelimumab and durvalumab, an anti-PD-L1 monoclonal antibody, as a first-line systemic therapy for advanced HCC, were first presented at the June 2022 American Society of Clinical Oncology annual meeting. In this study, patients were randomized to arms taking the single high priming dose of tremelimumab and durvalumab, an infusion regimen termed STRIDE (Single Tremelimumab Regular Interval Durvalumab), durvalumab monotherapy, or sorafenib monotherapy. The median OS was 16.43 months (95% CI, 14.16–19.58), 16.56 months (95% CI, 14.06–19.12), and 13.77 months (95% CI, 12.25–16.13), respectively, with an overall survival HR for STRIDE versus sorafenib of 0.78 (96% CI, 0.65–0.93; *p* < 0.0035) [54]. Additionally, of note is that durvalumab monotherapy was demonstrated to be non-inferior to sorafenib. Lastly, no new safety signals were noted. The regimen was approved by the US FDA in October 2022 and recommended as an additional first-line systemic therapy option for advanced HCC.
cancers-15-02506-t001_Table 1Table 1A summary of first-line systemic therapy options for HCC as outlined by NCCN guidelines.
RegimenTrial NameAuthorsYearStudy Arm# of PtsPrimary Endpoint Results“Preferred Regimens:”Atezolizumab (Checkpoint inhibitor: anti-PD-L1 monoclonal Ab) + Bevacizumab (anti-VEGF-A monoclonal Ab) combination therapyIMbrave150Finn et al. [51]2020Atezolizumab + bevacizumab vs. sorafenib (first-line setting) 501OS: at 12 months, 67.2% (95% CI, 61.3 to 73.1) with atezolizumab—bevacizumab and 54.6% (95% CI, 45.2 to 64.0) with sorafenib.mPFS: 6.8 months (95% CI, 5.7 to 8.3) and 4.3 months (95% CI, 4.0 to 5.6) in the respective groups (hazard ratio for disease progression or death, 0.59; 95% CI, 0.47 to 0.76; *p* < 0.001).
STRIDE (Tremelimumab (anti-CTLA-4 monoclonal Ab) + Durvalumab (checkpoint inhibitor: anti-PD-L1 monoclonal Ab)) combination therapyHIMALAYAAbou-Alfa et al. [54]2022STRIDE vs. durvalumab vs. sorafenib (first-line setting)1171OS: 16.4 months (14.2–19.6) w/STRIDE (single tremelimumab + regular interval durvalumab) and 13.8 months (12.3–16.1) with sorafenib (HR for death, 0.78; 96% CI, 0.65–0.92; *p* = 0.0035)“Other Recommended Regimens:”Sorafenib (TKI of VEGF-R1–3, PDGF beta and serine-threonine kinase inhibitor of Raf-1 and B-Raf) monotherapySHARPLovet et al. [8]2008Sorafenib vs. placebo (first-line setting) 602OS: 10.7 months with sorafenib and 7.9 months with placebo (HR, 0.69; 95% CI, 0.55 to 0.87; *p* < 0.001)mPFS: 4.1 months vs. 4.9 months (*p* = 0.77)
Lenvatinib (TKI of VEGF-R1–3, FGF receptors 1–4, PDGF receptor alpha, RET, KIT) monotherapyREFLECTKudo et al. [28]2018Lenvatinib vs. sorafenib (first-line setting)954OS: 13.6 months (95% CI, 12.1–14.9) for lenvatinib was non-inferior to 12.3 months (10.4–13.9; HR, 0.92; 95% CI, 0.79–1.06) for sorafenib 
Durvalumab (checkpoint inhibitor: anti-PD-L1 monoclonal Ab) monotherapy HIMALAYAAbou-Alfa et al. [54]2022STRIDE vs. durvalumab vs. sorafenib (first-line setting)1171OS: 16.6 months for durvalumab monotherapy vs. 13.8 months for sorafenib monotherapy (HR, 0.86; 95% CI, 0.73–1.03). Judged to be non-inferior. 
Pembrolizumab (checkpoint inhibitor: anti-PD-1 Monoclonal Ab) monotherapyKEYNOTE-224, 2021 update of Cohort 2Verset et al. [46]2022Phase II study of pembrolizumab 51ORR: 16% (95% CI, 7–29) for pembrolizumab monotherapy “Useful in Certain Circumstances:”Nivolumab (checkpoint inhibitor: anti-PD-1 monoclonal Ab) monotherapyCheckMate 459Yau et al. [47]2021Nivolumab vs. sorafenib (first-line setting) 743OS: 16.4 months (95% CI 13.9–18.4) with nivolumab and 14.7 months (11.9–17.2) with sorafenib (HR, 0.85; 95% CI, 0.72–1.02; *p* = 0.075)

#### 2.1.12. Other Trials Combining TKIs and ICIs

Around the same timeframe, the phase III COSMIC-312 trial compared the regimen of cabozantinib, a multikinase inhibitor (TKI) of kinases involved in tumor pathogenesis including VEGF, MET and the TAM family (TYRO3, AXL, MER), and the previously mentioned atezolizumab. This regimen was compared against sorafenib in the first-line setting for advanced HCC. The results of the dual predefined primary endpoints of PFS per RECIST 1.1 and OS were mixed. The median PFS was 6.8 months (99% CI, 5.6–8.3) in the combination treatment group versus 4.2 months (2.8–7.0) in the sorafenib group (HR, 0.63; 99% CI, 0.44–0.91; *p* = 0.0012). The median OS was 15.4 months (96% CI, 13.7–17.6) in the combination treatment group versus 15.5 months (12.1-not estimable) in the sorafenib group (HR, 0.90; 96% CI, 0.69–1.18; *p* = 0.44). Furthermore, serious treatment-related adverse events occurred in 18% of patients in the combination group (with five fatal events) compared to 8% in the sorafenib group (with one fatal event) [55]. Consequently, there was no formal adoption by the NCCN or a recommendation of this regimen. 

A phase Ib study combining lenvatinib and pembrolizumab in patients with unresectable HCC showed promising anti-tumor activity, with a mOS of 22 months, and no new identifiable safety signals [56]. In the ensuing phase III, global, randomized, double-blind LEAP-002 study, lenvatinib and pembrolizumab was compared to. lenvatinib monotherapy. Unfortunately, the primary endpoints of OS at the final analysis and PFS at an interim analysis I did not meet the pre-specified statistical significance [57]. 

### 2.2. Second-Line Systemic Therapies

Despite the fact that there are no comparative data with which to define optimal treatment after first-line systemic therapy, there currently exist 10 NCCN recommended regimens, with various degrees of US FDA approval for the treatment of advanced, progressed HCC in various circumstances. Notably, second-line therapy is an option for patients whose HCC progresses while in first-line therapy as well as for those patients whose presenting performance status and liver function are sufficient to tolerate it. The options are presented in Table 2. Importantly, the bulk of these studies were conducted on patients with disease progression on or after sorafenib monotherapy. 

#### 2.2.1. Approved Second-Line TKI Therapies

Regorafenib, a multikinase inhibitor targeting VEGFR 1–3, FGFR 1–2, angiopoietin-1 receptor (TIE2) and alpha and beta PDFRs, was the first agent to be approved as a second-line systemic therapy for advanced HCC following progression with sorafenib. In the RESORCE phase III randomized, double-blind, placebo-controlled trial, regorafenib demonstrated improved OS with a HR of 0.63 (95% CI, 0.50–0.79; one-sided *p* < 0.0001) and a mOS of 10.6 months (95% CI, 9.1–12.1) for regorafenib versus 7.8 months (6.3–8.8) for the placebo [58]. Notable adverse events that were shown to be increased in the regorafenib group were hypertension (15% to 5%), hand-foot-skin reaction (13% vs. 1%), fatigue (9% vs. 5%) and diarrhea (3% vs. 0). 

Cabozantinib, a multikinase inhibitor with unique activity against VEGFR 2, AXL, and MET was the next agent to improve efficacy outcomes versus the placebo in the second-line population who had received prior sorafenib irrespective of duration. In the CELESTIAL phase III clinical trial, cabozantinib was noted to improve OS relative to the placebo in the overall second-line population (mOS, 11.3 vs. 7.2 months; HR, 0.70; 95% CI, 0.55 to 0.88). This improvement was maintained in sub-analyses of the total primary treatment time of sorafenib with longer duration generally corresponding to longer median OS; the median OS of 8.9 vs. 6.9 months (HR, 0.72; 95% CI, 0.47 to 1.10) was for prior sorafenib use for <3 months, 11.5 vs. 6.5 months (HR, 0.65; 95% CI, 0.43 to 1.00) for use for 3 to <6 months and 12.3 vs. 9.2 months (HR, 0.82; 95% CI, 0.58 to 1.16) for use for ≥6 months [59]. Safety data were consistent with the overall study population, and thus cabozantinib entered the pool as an efficacious second-line systemic therapeutic option. 

#### 2.2.2. Second-Line Monoclonal Antibody Therapies

The randomized, double-blind, placebo-controlled, REACH-2 phase III clinical trial aimed to establish the efficacy of ramucirumab, a humanized monoclonal antibody directed against VEGF 2, in patients with advanced HCC, disease progression following first-line sorafenib therapy, and alpha-fetoprotein concentrations of 400 ng/mL or higher. In this category of patients, the mOS of the ramucirumab group was 8.5 months (95% CI, 7.0–10.6) vs. 7.3 months (5.4–9.1) for the placebo group (HR, 0.71; 95% CI, 0.531–0.949; *p* = 0.0199) [60]. Having reached its primary endpoint in showing improved OS with a manageable safety profile, ramucirumab was approved as a second-line systemic therapy for advanced HCC in patients with AFP > 400. 

As mentioned earlier, pembrolizumab was well studied in the KEYNOTE trials. KEYNOTE-240, a phase III randomized, double-blind clinical trial, sought to evaluate the efficacy and safety of pembrolizumab in previously treated patients with advanced HCC, building upon the noted anti-tumor activity and safety observed in the phase II KEYNOTE-224 trial. The primary endpoints were set as OS and PFS (with pre-defined one-sided significance thresholds, and *p* = 0.0174 for final analysis). The median OS was 13.9 months (95% CI, 11.6 to 16.0 months) for pembrolizumab versus 10.6 months (95% CI, 8.3 to 13.5 months) for the placebo (HR, 0.781; 95% CI, 0.611 to 0.998; *p* = 0.0238) [61]. The median PFS for pembrolizumab was 3.0 months (95% CI, 2.8 to 4.1 months) versus 2.8 months (95% CI, 1.6 to 3.0 months) at final analysis (HR, 0.718; 95% CI, 0.570 to 0.904; *p* = 0.0022). Grade 3 or higher adverse events occurred in 147 (52.7%) and 62 patients (46.3%) for pembrolizumab versus the placebo [61]. Although pembrolizumab did not reach the prespecified OS or PFS goals, there was a significant improvement in ORR (18.3% vs. 4.4%; nominal one-sided *p* = 0.0007) and pembrolizumab remained a US approved drug for both first- and second-line systemic therapy for advanced HCC. 

In CheckMate 040, an open-label, non-comparative, phase I/II dose-escalation and -expansion trial, nivolumab was investigated as a monotherapy in patients with advanced HCC with and without prior sorafenib treatment and/or progression. A total of 70% of patients studied in the different arms had undergone some form of treatment with sorafenib, thus allowing nivolumab to be examined as a second-line systemic therapy option. The ORR was 20% (95% CI, 15–26) in patients treated with 3 mg/kg nivolumab in the dose-expansion phase and 15% (95% CI, 6–28) in the dose-escalation phase [62]. This work paved the way for nivolumab adoption as a potential first- and second-line systemic option with the subsequent CheckMate 459 study evaluating more clearly its role in the first-line setting, as previously discussed. 

#### 2.2.3. Second-Line Combination Therapy: Nivolumab and Ipilimumab

An expansion arm of CheckMate 040 then examined the efficacy and safety of the combination regimen of nivolumab and ipilimumab, a humanized monoclonal CTLA-4 antibody. In this randomized three-arm, phase I/II clinical trial of 148 patients, nivolumab plus ipilimumab demonstrated manageable safety, promising ORR, and durable responses [63]. Patients were randomized at a ratio of 1:1:1 into three separate arms (A: 1 mg/kg nivolumab and 3 mg/kg ipilimumab, administered every 3 weeks (4 doses), followed by 240 mg nivolumab every 2 weeks; B: 3 mg/kg nivolumab and 1 mg/kg ipilimumab, administered every 3 weeks (4 doses), followed by 240 mg nivolumab every 2 weeks; C: 3 mg/kg nivolumab every 2 weeks and 1 mg/kg ipilimumab every 6 weeks) that differed in terms of both drug dosage and interval of administration of the regimen. In greater detail, ORR was observed to be 32% (95% CI, 20–47%) in arm A, 27% (95% CI, 15–41%) in arm B, and 29% (95% CI, 17–43%) in arm C [63]. Notably, any-grade treatment-related adverse events were observed in 46 patients (94%) in arm A, 35 patients (71%) in arm B, and 38 patients (79%) in arm C, and of the 10, 6, and 3 patients in the respective arms who had a hepatic immune-mediated adverse event, 7, 3, and 2 received high-dose glucocorticoids (≥40 mg of prednisone per day or equivalent) for a median of 2 weeks (0.9–7.0), 1 week (0.6–1.1), and 3 weeks (2.0–3.0), respectively [63]. Regardless, based on durable responses with high ORRs and a safety profile deemed manageable, the combination regimen received accelerated approval in the US as a second-line therapy for HCC. 

#### 2.2.4. Unsuccessful Second-Line Therapies

Tivantinib, a selective, oral MET inhibitor, was investigated in a phase III, randomized, double-blind, placebo-controlled study in 90 centers in Australia, the Americas, Europe, and New Zealand. The study was based on the foundational basic scientific knowledge of the hepatocyte growth factor (HGF), an autocrine and paracrine factor produced by stromal cells, inducing c-Met, a high-affinity tyrosine kinase receptor, thereby triggering a variety of cellular responses, including proliferation, survival, cytoskeleton rearrangements, cell–cell dissociation, and motility [64]. In cancer, research showed HGF/c-Met signaling to be a proliferative advantage, thus promoting tumor invasion and metastasis [65]. Early studies causally linked c-Met over-expression to poor prognosis in HCC; however, no evidence had yet indicated c-Met inhibition to be a viable treatment for HCC [66]. A phase I study examining the combination of the MET inhibitor, tivantinib, and sorafenib demonstrated a well-tolerated safety profile [67]. A follow-up phase II study demonstrated the efficacy and safety of tivantinib for the second-line treatment of advanced HCC, particularly for patients with tumors with high level of MET amplification [68]. In the phase III trial, however, at a median follow up of 18.1 months, the mOS was 8.4 months (95% CI, 6.8–10.0) in the tivantinib group and 9.1 months (7.3–10.4) in the placebo group (HR, 0.97; 95% CI, 0.75–1.25; *p* = 0.81) [69]. It was consequently established that tivantinib did not improve overall survival compared with placebo in patients with high level, MET-amplified, advanced HCC previously treated with sorafenib. 

Similarly, the mitogen-activated protein kinase pathway, also known as the RAS/RAF/MEK/extracellular signaling kinase (ERK) (MAPK pathway) was investigated as a potential source of inhibition in advanced HCC given its ubiquitous intracellular signaling transduction role [70]. Unlike in other solid cancers, mutations in the RAS and RAF genes are rarely found in HCC, and instead there is an overexpression of MEK and ERK as the mechanisms of MAPK pathway activation in HCC [71,72,73]. Early studies demonstrated that the allosteric MEK1/2 inhibitor, refametinib, in combination with sorafenib exhibits anti-tumor activity in preclinical murine and rat models of HCC, thus supporting ongoing research [74]. A subsequent phase I study demonstrated refametinib and sorafenib to be well tolerated, with good oral absorption, near-dose proportionality, and target inhibition in a range of tumor types, including HCC [75]. In a phase II study examining refametinib in HCC patients, all patients were screened for KRAS and then separated into refametinib monotherapy or refametinib–sorafenib combination therapy groups. The radiological response rate was 0% in the monotherapy arm and 6.3% in the combination arm, and the mOS was 12.7 months in the latter group [76]. As a result of low response rates and less than impressive survival results, there was no further progression to the phase III setting and further development of refametinib was curtailed.
cancers-15-02506-t002_Table 2Table 2A summary of second-line systemic therapy options for HCC as outlined by NCCN guidelines.
RegimenTrial NameAuthorsYearStudy Arm# of PtsPrimary Endpoint Results“Preferred Regimens:”Regorafenib (a multikinase inhibitor targeting VEGFR 1–3, FGFR 1–2, angiopoietin-1 receptor (TIE2) and PDFRs alpha and beta)RESORCEBruix et al. [58]2017Regorafenib vs. placebo (second-line setting) 573OS: 10.6 months (95% CI 9.1–12.1) for regorafenib versus 7.8 months (6.3–8.8) for placebo (HR of 0.63; 95% CI, 0.50–0.79; one-sided *p* < 0·0001)
Cabozantinib (a multikinase inhibitor (TKI) of kinases involved in tumor pathogenesis including VEGF, MET and the TAM family (TYRO3, AXL, MER)CELESTIALKelley et al. [59]2020Cabozantinib vs. placebo (second-line setting)707OS: Cabozantinib improved OS relative to placebo in the overall second-line population who had received only prior sorafenib (median 11.3 vs. 7.2 months; HR, 0.70; 95% CI, 0.55 to 0.88)
Ramucirumab (humanized monoclonal antibody directed against VEGF 2) monotherapyREACH-2Zhu et al. [60]2019Ramucirumab vs. placebo (second-line setting) 292OS: At a median follow-up of 7.6 months (IQR 4.0–12.5), mOS was 8.5 months (95% CI 7.0–10.6) in the ramucirumab group vs. 7.3 months (5.4–9.1) in the placebo group (HR, 0.710; 95% CI, 0.531–0.949; *p* = 0.0199)
Lenvatinib (TKI of VEGF-R1–3, FGF receptors 1–4, PDGF receptor alpha, RET, and KIT) monotherapyREFLECTKudo et al. [28]2018Lenvatinib vs. sorafenib (first-line setting)954OS: 13.6 months (95% CI, 12.1–14.9) for Lenvatinib was non-inferior to 12.3 months (10.4–13.9; HR, 0.92; 95% CI 0.79–1.06) for sorafenib 
Sorafenib (TKI of VEGF-R1–3, PDGF beta and serine-threonine kinase inhibitor of Raf-1 and B-Raf) monotherapySHARPLovet et al. [8]2008Sorafenib vs. placebo (first-line setting) 602OS: 10.7 months with sorafenib and 7.9 months with placebo; HR, 0.69; 95% CI, 0.55 to 0.87; *p* < 0.001mPFS: 4.1 months vs. 4.9 months; *p* = 0.77“Other Recommended Regimens:”Nivolumab (checkpoint inhibitor: anti-PD-1 monoclonal antibody) + Ipilimumab (anti-CTLA-4 humanized monoclonal antibody)CheckMate 040Yau et al. [63]2020Phase I/II, three-arm study of nivolumab and ipilimumab (second-line setting) 148ORR: 32% (95% CI, 20–47%) in arm A, 27% (95% CI, 15–41%) in arm B, and 29% (95% CI, 17–43%) in arm C, with the respective arms differing in quantity and time for the administration of drugs 
Pembrolizumab (checkpoint inhibitor: anti-PD-1 monoclonal antibody monotherapyKeynote-240Finn et al. [61]2020Pembrolizumab vs. placebo (second-line setting) 413OS: 13.9 months (95% CI, 11.6 to 16.0 months) for pembrolizumab versus 10.6 months (95% CI, 8.3 to 13.5 months) for placebo (HR, 0.781; 95% CI, 0.611 to 0.998; *p* = 0.0238).mPFS (with predefined one-sided significance thresholds; *p* = 0.0174 for final analysis) for pembrolizumab was 3.0 months (95% CI, 2.8 to 4.1 months) versus 2.8 months (95% CI, 1.6 to 3.0 months) at final analysis (HR, 0.718; 95% CI, 0.570 to 0.904; *p* = 0.0022).“Useful in Certain Circumstances:”Nivolumab monotherapy CheckMate 040El-Khoueiry et al. [62]2017Phase I/II dose escalation and expansion trial assessing safety and efficacy of nivolumab monotherapy262ORR: 20% (95% CI 15–26) in patients treated with nivolumab 3 mg/kg in the dose-expansion phase and 15% (95% CI 6–28) in the dose-escalation phase
Dostarlimab-gxly (humanized anti-PD-1 monoclonal antibody) monotherapy GARNETAndre et al. [77]N/aPhase I study evaluating safety of dostarlimabN/aN/a
Selpercatinib (highly selective RET kinase inhibitor) monotherapyLIBRETTO-001Subbiah et al. [78]2022Phase ½ study evaluating safety and efficacy of selpercatinib in RET fusion-positive advanced solid non-lung or thyroid tumorsN/aN/a

#### 2.2.5. For Advanced, Refractory HCC Patients with MSI-H/dMMR Tumors

The ongoing phase I GARNET study (NCT02715284) is evaluating dostarlimab, a humanized anti-PD-1 monoclonal antibody, in patients with advanced solid tumors. In one specific cohort (cohort F) of this study, patients were enrolled with dMMR or POLE mutations of their non-endometrial solid tumors. It is too early to draw conclusions from this trial; however, pembrolizumab and nivolumab are approved for MSI-H of any primary site and can be used [77]. 

#### 2.2.6. For RET Gene Fusion-Positive Tumors

Similarly, the LIBRETTO-001 open-label, phase I/II, basket trial is underway, investigating the efficacy and safety of selpercatinib, a highly selective RET kinase inhibitor with CNS activity that has shown efficacy in RET fusion-positive lung and thyroid cancers, in a group of patients with RET fusion-positive advanced solid non-lung or thyroid tumors [78]. Despite early indications of clinically meaningful activity in the RET fusion-positive tumor population and a safety profile consistent with that observed in other indications, there were no patients enrolled with HCC. Thus, again, no conclusions can be drawn other than that of an additional clinical trial option for patients within this category. 

## 3. Our Current Approach to Systemic Therapy for HCC (or “How We Treat HCC”)

We now have category 1 evidence for two combination approaches in advanced HCC, including atezolizumab with bevacizumab and durvalumab with tremelimumab, and these offer a choice to clinicians [79]. These approaches appear to be well tolerated and to offer the hope (which is common in immune therapies for other cancers) that long term remissions may be achieved in a subset of patients. Similarly, there are several other approved therapeutic options for first- and second-line therapies. Finally, there is an emerging role of targeted therapies based on the specific molecular features of HCC. As a result of the rapid development of these options, challenges have arisen for clinicians in terms of the appropriate modalities to offer patients. This difficulty has been amplified due to the lack of head-to-head comparative studies between respective first-line and second-line regimens. In efforts to dispel this confusion, Sonbol et al., via a network meta-analysis (NMA) of 14 clinical trials mentioned in this review, concluded that atezolizumab with bevacizumab was superior in the first-line setting compared to sorafenib, lenvatinib, and nivolumab [80]. For second line studies, the OS benefit was only seen with regorafenib (HR, 0.62; 95% CI, 0.51–0.75) and cabozantinib (HR, 0.76; 95% CI, 0.63–0.92) compared with the placebo, as well as ramucirumab when accounting for patients with α-fetoprotein (AFP) levels of 400 ng/mL or greater [80]. 

It is now clear that conducting a biopsy, when feasible, is important for determining the optimal choice of therapy. HCC is one of the few cancers that has been reliably diagnosed based on imaging criteria and clinical features alone. This approach still has potential benefits in the setting of patients amenable to a resection or transplant for a cure since there is a discrete risk of tracking tumor cells through percutaneous biopsies. However, for more advanced patients, we need to exclude combined HCC and cholangiocarcinoma, which remains an area of unmet therapeutic need (being excluded in general from trials for HCC or cholangiocarcinoma). In addition, the emergence of targeted approaches for MSI-high HCC or HCC with NTRK fusion or RET mutations (or other targets yet to be studied) highlights the importance of biopsies. Blood-based tumor DNA assays are likely to also become helpful in HCC treatment in the future. It is possible that blood assays of other tissues may help to distinguish patients who are more likely to respond to immunotherapy vs. those more likely to respond to combined therapy approaches. 

The increasing effectiveness of systemic therapies means that these approaches should be moved up in the sequence of treatments rather than waiting until all interventional maneuvers have been exhausted. There remain, however, some major areas of unmet need. A major limitation of the atezolizumab and bevacizumab regimen is the risk of bleeding and arterial events with the latter agent [51]. Varices must be carefully evaluated and treated and the risk of cardiac or other arterial events must be assessed. Onco-cardiologic consultation may be helpful in select cases. If these contraindicate the use of bevacizumab then the durvalumab and tremelimumab regimen should be considered. One very attractive feature of the latter approach is that a single priming dose of the CTLA4 inhibitor (tremelimumab) is used in contrast to other combined immunotherapy approaches (e.g., nivolumab and ipilimumab) [54]. This may reduce the risk of major immuno-toxicity compared to other combined regimens (although notably, immune related adverse events were more frequent with the STRIDE combined regimen than durvalumab alone in the HIMALAYA trial). 

The risks of any immunotherapy must be weighed carefully in the setting of auto-immune disease or if there is a possibility of liver transplant in the future due to concerns that prior treatment with immunotherapy could lead to transplant rejection. If immunotherapy is contraindicated, then the major first line choices are between sorafenib and lenvatinib. Given the evidence of the lack of benefit of sorafenib in patients whose HCC is the result of hepatitis B infection, lenvatinib is the preferred choice for those patients. If combined immunotherapy seems too risky for any reason, then durvalumab alone is an option in the first line since it was non-inferior to sorafenib alone in the HIMALAYA trial. Note also that nivolumab has been tested in the setting of Child’s B cirrhosis and found to be safe [81], so this is an option for patients deemed unsafe for the use of TKIs due to the level of liver dysfunction. 

It is of note that the doses of lenvatinib used in HCC (i.e., 12 or 8 mg per day based on body weight) are lower than those used in some other cancers (e.g., 24 mg per day in thyroid cancer) [82]. In addition, it has been shown that use of lower doses of sorafenib when necessary (i.e., due to the severity of hand–foot syndrome or other complications) can have similar benefits to the those of taking a full dose of 400 mg twice daily [83]. Lenvatinib has the advantage of a higher response rate than sorafenib and a lower incidence of hand–foot syndrome. One caution for lenvatinib is the risk of rapid acceleration of blood pressure in some patients. This is a complication of sorafenib and other agents that inhibit VEGF signaling but appears to be more pronounced and rapid with lenvatinib [28]. On the other hand, if hand–foot syndrome is a major concern, lenvatinib is preferred over sorafenib. If a patient progresses on sorafenib, then second-line options include regorafinib, cabozantinib, and ramucirumab (if AFP levels are at 400 ng/mL or greater) due to the OS benefits stated above. If the patient receives single-agent lenvatinib or sorafenib but is then found to be able to tolerate immunotherapy, either single-agent nivolumab, pembrolizumab or combined nivolumab and ipilimumab are options. 

## 4. Future Directions

There are numerous ongoing and proposed clinical trials offering novel targeted therapy strategies. These trials include similar approaches to the previously mentioned molecular targets and innovative methods of treatment delivery. We expect significant further progress in the dynamic field of therapy for HCC. As molecular analysis of HCC tumors becomes more frequently performed, the identification of subsets amenable to specific targeted therapies can also be expected. It is possible also that HCC resulting from different underlying causes (e.g., steatohepatitis) may respond more or less to specific treatment approaches. As noted, sorafenib appears to be of more benefit in HCC arising from hepatitis C than that arising from hepatitis B infection. 

Currently, clinical trials that are not yet FDA-approved and/or ongoing consist of the following.

### 4.1. Novel VEGF Monotherapies and Combination Therapies

The DEDUCTIVE trial, a phase Ib/II, open-label study of tivozanib in combination with durvalumab in subjects with advanced hepatocellular carcinoma, first presented during the 2022 Gastrointestinal Cancers Symposium, revealed promising results. The trial was composed of two cohorts: A, which included patients with previously untreated HCC (and thus presenting a potential first-line treatment option), and B, which examined the combination therapy in patients with HCC who were pretreated with bevacizumab and atezolizumab. Tivozanib, a potent and selective VEGFR 1, 2 and 3 tyrosine-kinase inhibitor, had previously demonstrated anti-neoplastic efficacy as a monotherapy, as evident through the ORR of 21% vs. 18.8% (lenvatinib) and 6.5% (sorafenib) with similar safety profiles [84]. Despite failing to meet the threshold to activate stage 2, investigators hypothesized that the selectivity and favorable tolerability of tivozanib would allow it to be leveraged as part of a combination with an immune checkpoint inhibitor, thus was born the DEDUCTIVE trial. In preliminary results from 18 evaluable patients in Cohort A, tivozanib and durvalumab elicited an ORR of 27.8%, a mPFS of 7.3 months (95% CI, 1.8—not evaluable), and a mOS of 13.4 months (95% CI, 8.2—not evaluable) [85]. The estimated study completion date is March 2023 with hope for a phase III clinical trial to compare the efficacy to that of an established first line. Concomitantly, cohort B is still being enrolled with yet-to-be-examined data as a second-line option. 

### 4.2. Approved Systemic Therapies in China

Two agents approved in China, donafenib, a novel multikinase inhibitor and a deuterated sorafenib derivative, which showed superiority in mOS compared to sorafenib (12.1 vs. 10.3 months (HR, 0.831; *p* = 0.0363) in first line treatment [86] and apatinib, an orally active VEGFR-2 inhibitor, approved in second line [87], have not been evaluated elsewhere as of yet. They represent potential future agents pending US regulatory review and/or future studies. 

### 4.3. Novel Targeted Therapies and ICIs in Development

Current additional relevant targets for HCC progression prevention consist of the TGF-beta signaling pathway, fibroblast growth factor receptor 4 (FGFR4) and its associated growth factors, and colony-stimulating factor 1 (CSF-1). In addition to nivolumab and pembrolizumab, several alternative PD-1 inhibitors and PD-L1 inhibitors are currently being investigated in the clinical trial pipelines. Pertinently, the RATIONALE 301 phase III trial, first presented at the 2022 European Society for Medical Oncology Congress, is noted to have met its final analysis objective, demonstrating non-inferior overall survival (OS) for tislelizumab vs. sorafenib (mOS 15.9 months vs. 14.1 months; HR, 0.85; 95% CI 0.712–1.019) in patients with previously untreated unresectable hepatocellular carcinoma [88]. Zhang et al. recently effectively highlighted the total breadth of current clinical trials [89]. 

### 4.4. Alternative Combination Regimens

The Launch Trial, a phase III study, examined lenvatinib and TACE (trans arterial chemoembolization) compared with lenvatinib monotherapy. Limitations, such as the marked heterogeneity of the BCLC stage C population in terms of tumor distribution and disease burden, the younger age of the population, and the high prevalence of hepatitis B etiology, to name a few, are apparent. Despite these, the TACE and lenvatinib group demonstrated a mOS (the primary endpoint) of 17.8 months vs. 11.5 months with the lenvatinib alone group (HR, 0.45; 95% CI 0.33–0.61; *p* < 0.001), leading to a dramatic 55% reduction in the risk of death [90]. Thus, while applicability concerns certainly exist, there remain clear and present data to support its role in further research into bringing back liver-directed therapy in some treatment capacity. 

In a similar multimodal frame, the phase III NRG/RTOG 1112 trial assessed sorafenib combined with stereotactic body radiation (SBRT) compared to sorafenib alone. According to data presented at the 2022 American Society for Radiation Oncology Annual Meeting (abstract LBA01), mOS, the primary endpoint, was 12.3 months (90% CI, 10.6–14.3) in the sorafenib arm vs. 15.8 months (90% CI, 11.4–19.2) with the addition of SBRT (HR, 0.77; 90% CI, 0.59–1.01; *p* = 0.55) [91]. A secondary endpoint, the mPFS was 5.5 months (95% CI, 3.4–6.3) for patients receiving sorafenib monotherapy and 9.2 months (95% CI, 7.5–11.9) for those receiving sorafenib with SBRT (HR, 0.55; 95% CI, 0.40–0.75; *p* = 0.0001). One aspect of particular interest for this trial was the inclusion of patients with macrovascular invasion, excluded in the HIMALAYA trial (tremelimumab and durvalumab), but not the Imbrave150 trial (atezolizumab and bevacizumab). Further studies combining SBRT with combined systemic therapies may be of interest.

Since both trials were initiated prior to the current standard use of combined systemic therapy, it is unclear how they will fit into treatment paradigms apart from settings where single-agent tyrosine kinase inhibitors are opted for. Combining systemic agents with radio-embolization (e.g., with Yttrium 90) or combining radiation with immunotherapy are also of interest. At present, there is no proven adjuvant therapy for patients who have undergone surgery or definitive ablation procedures. An adjuvant trial of sorafenib was negative [92]. Ongoing trials of immunotherapy in this setting are awaited with interest. 

### 4.5. Chemoprevention of HCC with Non-Antitumoral Agents

Various studies have evaluated the effects of the commonly prescribed lipid-lowering medication, statins, on the risk of developing HCC, the premise being that statins, or 3-hydroxy-3-methylglutaryl coenzyme (HMG-CoA) reductase inhibitors, have shown that in addition to cholesterol reduction, have antiproliferative, proapoptotic, antiangiogenic, immunomodulatory, and anti-infective effects, thus preventing cancer growth [93]. A recent meta-analysis, with a primary endpoint of time-dependent correlation between statin use and HCC incidence expressed as HR, demonstrated a crude odds ratio (OR) for a HCC incidence of 0.59 (95% CI, 0.47–0.74), later confirmed in an adjusted analysis (OR, 0.74; 95% CI, 0.70–0.78) [94]. These results show the beneficial chemo-preventative effect of statins against HCC occurrence, when administered in a dose-dependent manner, and that they are more pronounced in lipophilic varieties. As these studies were prone to selection bias given the inclusion of a majority of retrospective series, in addition to other limitations, further confirmatory clinical studies are warranted. The importance of these findings may play a role from the primary prevention of HCC up to tertiary prevention coinciding with systemic therapy options. 

### 4.6. Emerging Treatment Strategies

Blocking HIF1alpha (as has been shown to have benefits in renal cell carcinoma) may prove valuable in treating HCC. Other novel approaches derived from the treatment of hematological cancers include chimeric antigen receptor T cells (CAR-T) and bispecific T -cell engagers (BiTE). For CAR T-cells, early studies have investigated glypican (GPC3), a cell-surface glycophosphatidylinositol (GPI)-anchored protein that belongs to the heparan sulfate (HS) proteoglycan family, which plays important roles in cell growth differentiation, and migration, and is noted to be disproportionately elevated in HCC [95]. Thus, GPC3 served as a natural starting point for CAR T-cell therapy. Early, phase I trial data have demonstrated the initial safety profile of CAR-glypican-3 T-cell therapy with early signs of antitumor activity in patients with advanced HCC [96]. Currently, CAR T-cells against other antigens of previously mentioned relevance to HCC are under investigation in ongoing clinical trials, notably anti-c-MET/PD-L1 (NCT03672305). In foundational BiTE studies, GPC3 has similarly served as a starting point for targeting. Thus far, ERY974, a humanized IgG4 bispecific T-cell-redirecting antibody recognizing GPC3 and CD3, was investigated in a phase I trial in which an appropriate dosage was administered without any adverse events [97]. As a final note, the introduction of agents acting through mechanisms other than VEGF inhibition or immune enhancement could be helpful for patients who have contra-indications to either approach (e.g., those with prior transplants and cardiovascular or bleeding risks). 

## 5. Conclusions

HCC is a highly prevalent and fatal cancer that accounts for a staggering amount of morbidity and mortality throughout the world. Despite the improvements in surveillance screening programs, the prevalence of HCC is expected to rise significantly over the coming decades. Furthermore, even with advancements in the technical capabilities of treatments for early-presenting HCC, most patients with HCC present with (or ultimately progress toward) advanced stages, thus placing importance on developing effective systemic therapy options. It is hoped that preventive approaches including using effective treatments (or prevention strategies) for hepatis B and C and addressing the rising incidence of steatohepatitis will have major benefits for reducing the incidence of HCC. Beginning with the approval of sorafenib in 2007, there has been a monumental amount of progress attained in the management of advanced HCC with systemic therapies, and the current pace of discovery and innovation promises further benefits to patients. Certain major gaps in therapy remain. A major treatment gap exists for patients with more advanced cirrhosis. The HIMALAYA trial only included patients with Child’s A to B7 cirrhosis. The combined atezolizumab and bevacizumab trial did not include any patients with Child’s B-stage cirrhosis. In addition, new advances in early diagnosis and the treatment of HCC are not reaching underserved populations as racial disparities in care continue to exist within the United States and the expensive new therapies have limited global reach [98]. Future directions promise new, innovative ways to target advanced HCC. Ongoing clinical trials and those to come continue to bring hope to a once-stagnant treatment field.

## Figures and Tables

**Figure 1 cancers-15-02506-f001:**
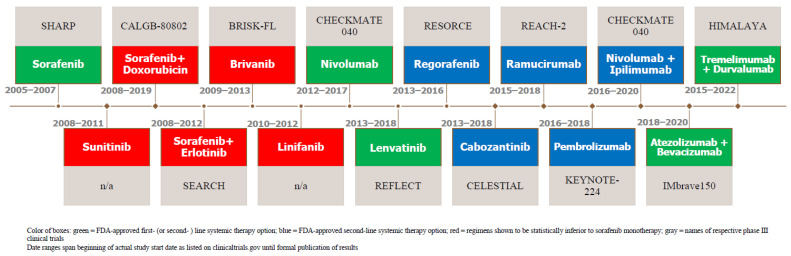
Timeline of Notable HCC Systemic Therapy Trials.

## Data Availability

No new data were created or analyzed in this study. Data sharing is not applicable to this article.

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
