# Peer review of "A Comprehensive Narrative Review on the History, Current Landscape, and Future Directions of Hepatocellular Carcinoma (HCC) Systemic Therapy"

_cancers, 2023, doi:10.3390/cancers15092506_

Round 1
Reviewer 1 Report
Very well written and comprehensive (the title is right!) review! My comments:
1) While the current reviewer appreciates the historical and "anecdotal" value of the "prelude", the manuscript is too long and hard to read in the current version. I would recommend to delete this section.
2) In the evolution of systemic therapy, the authors should comment also some agents tested unsuccessfully in the past such as tivantinib or MEK 1/2 inhibitors.
3) When describing immunotherapy, the authors should focus on the safety profile with particular interest to the immune-mediated adverse events such as reactivation of IBD (cite the recent meta-analysis on this topic)
4) The authors should add some comments on combination therapies, i.e. combo therapies between systemic agents and loco-regional treatments
5) Also the section "future direction" is too long and speculative....i would focus mainly on current evidence....
6) The authors should comment on the potential prognostic role of other drugs (not antitumoral agents) in these patients (cite the recent MA PMID: 32260179)
It is OK
Author Response
Thank you for your time, consideration and feedback. We have addressed the individual points below.
1. Understood. We value your appreciation for the context but recognize it differs from the norm. We deleted the entire section from the paper.
2. We agree with your input. The following sub-section has been added to Section II. Evolution of Systemic Therapy:
Unsuccessful Second-line Therapies
Tivantinib, a selective, oral MET inhibitor, was investigated in a phase III, randomized, double-blind, placebo-controlled study in 90 centers in Australia, the Americas, Europe, and New Zealand. The study was based on the foundational basic science knowledge of hepatocyte growth factor (HGF), an autocrine and paracrine factor produced by stromal cells, inducing c-Met, a high affinity tyrosine kinase receptor, thereby triggering a variety of cellular responses, including proliferation, survival, cytoskeleton rearrangements, cell-cell dissociation, and motility [64]. In cancer, research had shown HGH/c-Met signaling to be a proliferative advantage, thus promoting tumor invasion and metastasis [65]. Early studies causally linked c-Met over-expression to poor prognosis in HCC, however no evidence had yet indicated c-Met inhibition to be a viable treatment for HCC [66]. A phase I study examining the combination of the MET inhibitor, tivantinib, plus sorafenib demonstrated a well-tolerated safety profile [67]. A follow-up phase II study demonstrated efficacy and safety of tivantinib for second-line treatment of advanced HCC, particularly for patients with MET-high tumors [68]. In the phase III trial however, at a median follow up of 18.1 months, mOS was 8.4 months (95% CI, 6.8-10.0) in the tivantinib group and 9.1 months (7.3-10.4) in the placebo group (HR, 0.97; 95% CI, 0.75-1.25; p=0.81) [69]. It was consequently established that tivantinib did not improve overall survival compared with placebo in patients with MET-high advanced HCC previously treated with sorafenib.
Similarly, the mitogen-activated protein kinase pathway, also known as the RAS/RAF/MEK/extracellular signaling kinase (ERK) [MAPK pathway] was investigated as a potential source of inhibition in advanced HCC given its ubiquitous intracellular signaling transduction role [70]. Unlike other solid cancers, mutations in the RAS and RAF genes are rarely found in HCC, and instead there is overexpression of MEK and ERK as the mechanisms of MAPK pathway activation in HCC [71-73]. Early studies demonstrated that the allosteric MEK1/2 inhibitor, refametinib, in combination with sorafenib exhibits antitumor activity in preclinical murine and rat models of HCC, thus supporting ongoing research [74]. A subsequent phase I study demonstrated refametinib plus sorafenib to be well tolerated, with good oral absorption, near-dose proportionality, and target inhibition in a range of tumor types, including HCC [75]. In a phase II study examining refametinib in HCC patients, all patients were screened for KRAS and then separated into refametinib monotherapy or refametinib-sorafenib combination therapy groups. The radiological response rate was 0% in the monotherapy arm and 6.3% in the combination arm, and the mOS was 12.7 months in the latter group [76]. As a result of low response rates and less than impressive survival results, there was no further progression to the phase III setting and further development of refametinib was curtailed.
3. We agree with your input. The following paragraph was added on pages 7-8 under the sub-section of Immune Checkpoint Inhibitors (ICI): Origin in Treatment for Advanced HCC:
Despite the clear clinical benefits achieved with ICI therapy thus far, it is worth noting their use has been associated with a new spectrum of side effects, related to their mechanism of action, which is distinct from other systemic therapies such as cytotoxic chemotherapy [41]. Notably, their side effects predominantly effect the GI, dermatologic, hepatic, endocrine, and pulmonary systems though any bodily system may be affected. The incidence and onset of immune-related adverse effects has been shown to depend on the type of cancer, the class and dose of ICI used, and specific factors related to the patient [41]. Because HCC by and large develops on a background of cirrhosis, which on its own leads to systemic manifestations, the potential for harmful risks could not be ignored. However, in general, studies have shown that metrics for immune related adverse events are not significantly higher for HCC treatment than they are for other cancers [42]. Certain patient populations with specific co-morbidities must be more carefully examined when undergoing ICI therapy, as is evident in those with IBD (inflammatory bowel disease) undergoing disease reactivation at higher rates [43].
4. We respect your input but believe that the point was addressed in the section Alternative Combination Regimens:
The Launch Trial, a phase III study, examined lenvatinib plus TACE (trans arterial chemoembolization) compared with lenvatinib monotherapy. Limitations such as the marked heterogeneity of the BCLC stage C population in terms of tumor distribution and disease burden, younger age of population, and high prevalence of hepatitis B etiology to name a few are apparent. Despite these, the TACE plus lenvatinib group demonstrated a mOS (the primary endpoint) of 17.8 months vs 11.5 months with the lenvatinib alone group (HR, 0.45; 95% CI 0.33-0.61; p<0.001), leading to a dramatic 55% reduction in the risk of death [90]. Thus, while applicability concerns certainly exist, there remains clear and present data to support a role for further research into bringing back liver-directed therapy in some treatment capacity.
In a similar multi-modal frame, the phase III NRG/RTOG 1112 trial assessed sorafenib combined with stereotactic body radiation (SBRT) compared to sorafenib alone. According to data presented at the 2022 American Society for Radiation Oncology Annual Meeting (abstract LBA01), mOS, the primary endpoint, was 12.3 months (90% CI, 10.6-14.3) in the sorafenib arm vs 15.8 months (90% CI, 11.4-19.2) with the addition of SBRT (HR, 0.77; 90% CI, 0.59-1.01; p=0.55) [91]. A secondary endpoint, the mPFS was 5.5 months (95% CI, 3.4-6.3) for patients receiving sorafenib monotherapy and 9.2 months (95% CI, 7.5-11.9) for those receiving sorafenib with SBRT (HR, 0.55; 95% CI, 0.40-0.75; p=0.0001). One aspect of particular interest for this trial was inclusion of patients with macrovascular invasion, excluded in the HIMALAYA trial (tremelimumab and durvalumab), but not the Imbrave150 trial (atezolizumab and bevacizumab). Further studies combining SBRT with combined systemic therapies may be of interest.
Since both trials were initiated prior to current standard use of combined systemic therapy, it is unclear how they will fit into treatment paradigms apart from settings where single agent tyrosine kinase inhibitors are opted for. Combining systemic agents with radio-embolization (e.g., with Yttrium 90) or combining radiation with immunotherapy are also of interest. At present there is no proven adjuvant therapy for patients who have undergone surgery or definitive ablation procedures. An adjuvant trial of sorafenib was negative [92]. Ongoing trials of immunotherapy in this setting are awaited with interest.
5. Point is well-taken. We reduced the section regarding agents studied in China but not yet approved in the US. We do however, feel the mentioning of CAR T cells and BiTE studies under Emerging Treatment Strategies is worthy of being mentioned as a future systemic therapy option. Technology has thus far shown promise in hematologic cancers and it is not entirely speculative to the point of unfeasibility.
6. We appreciate this suggestion. We have added the following sub-section to the Future Direction Section:
Chemoprevention of HCC with Non-antitumoral Agents
Various studies have evaluated the effects of the commonly prescribed lipid-lowering medication, statins, on the risk of developing HCC. The premise being that statins, or 3-hydroxy-3-methylglutaryl coenzyme (HMG-CoA) reductase inhibitors, have shown that in addition to cholesterol reduction, they have antiproliferative, proapoptotic, antiangiogenic, immunomodulatory, and anti-infective effects, thus preventing cancer growth [93]. A recent meta-analysis, with a primary endpoint of time-dependent correlation between statin use and HCC incidence expressed as HR, demonstrated a crude odds ratio (OR) for HCC incidence of 0.59 (95% CI, 0.47-0.74), later confirmed in adjusted analysis (OR, 0.74; 95% CI, 0.70-0.78) [94]. These results show a beneficial chemo preventative effect of statins against HCC occurrence, in a dose-dependent manner and more pronounced in lipophilic varieties. Prone to selection bias given the inclusion of majority retrospective series, in addition to other limitations, further confirmatory clinical studies are warranted. The importance of these findings may play a role in primary prevention of HCC up to that of tertiary prevention coinciding with systemic therapy options.
Reviewer 2 Report
This is a review article focusing on systemic therapy for hepatocellular carcinoma. I have several comments.
1.The Prelude section is not commonly used in most articles.
2. In page 17, what is "arm A", "arm B", and "cohor F"?
3. Why did the authors compare with renal cell carcinoma?
Author Response
Thank you for your time, consideration, and valuable input. We have addressed your points individually below.
1. We recognize this is not commonly used and attempted to provide a unique perspective. That said, we understand and have opted to remove the entire section from the article.
2. We have added the following sentence in the paragraph to better explain the three arms of the study:
Patients were randomized 1:1:1 into three separate arms [A, nivolumab 1 mg/kg plus ipilimumab 3 mg/kg, administered every 3 weeks (4 doses), followed by nivolumab 240 mg every 2 weeks; B, nivolumab 3 mg/kg plus ipilimumab 1 mg/kg, administered every 3 weeks (4 doses), followed by nivolumab 240 mg every 2 weeks; C, nivolumab 3 mg/kg every 2 weeks plus ipilimumab 1 mg/kg every 6 weeks] that differed on both drug dosage and interval of administration of the regimen.
As for cohort F, we have changed the language ever so slightly to simply make the point that a select group of patients in a single cohort of the study contains the dMMR or POLE mutations. It now reads as follows:
In one specific cohort (cohort F) of this study, patients were enrolled with dMMR or POLE mutations of their non-endometrial solid tumors.
3. We were attempting to draw causal links between the two due to the similarities in ridding the body of toxins and difficulties treating with traditional chemotherapy. That said, taking advice from you and fellow reviewer, we have decided to remove the discussion as it appears it detracts from the focus on HCC systemic therapy.
Round 2
Reviewer 1 Report
The revised version is OK. Thank you!
Reviewer 2 Report
The authors have revised the manuscript appropriately.